# Drug Delivery Strategies for Curcumin and Other Natural Nrf2 Modulators of Oxidative Stress-Related Diseases

**DOI:** 10.3390/pharmaceutics13122137

**Published:** 2021-12-12

**Authors:** Nina Katarina Grilc, Matej Sova, Julijana Kristl

**Affiliations:** 1Department of Pharmaceutical Technology, Faculty of Pharmacy, University of Ljubljana, Aškerčeva 7, 1000 Ljubljana, Slovenia; Nina.Katarina.Grilc@ffa.uni-lj.si; 2Department of Pharmaceutical Chemistry, Faculty of Pharmacy, University of Ljubljana, Aškerčeva 7, 1000 Ljubljana, Slovenia; matej.sova@ffa.uni-lj.si

**Keywords:** Nrf2 modulator, oxidative stress, oral bioavailability, poor solubility, curcumin, resveratrol, solid dispersions, SMEDDS, nanoparticles, micelles

## Abstract

Oxidative stress is associated with a wide range of diseases characterised by oxidant-mediated disturbances of various signalling pathways and cellular damage. The only effective strategy for the prevention of cellular damage is to limit the production of oxidants and support their efficient removal. The implication of the nuclear factor erythroid 2-related factor 2 (Nrf2) pathway in the cellular redox status has spurred new interest in the use of its natural modulators (e.g., curcumin, resveratrol). Unfortunately, most natural Nrf2 modulators are poorly soluble and show extensive pre-systemic metabolism, low oral bioavailability, and rapid elimination, which necessitates formulation strategies to circumvent these limitations. This paper provides a brief introduction on the cellular and molecular mechanisms involved in Nrf2 modulation and an overview of commonly studied formulations for the improvement of oral bioavailability and in vivo pharmacokinetics of Nrf2 modulators. Some formulations that have also been studied in vivo are discussed, including solid dispersions, self-microemulsifying drug delivery systems, and nanotechnology approaches, such as polymeric and solid lipid nanoparticles, nanocrystals, and micelles. Lastly, brief considerations of nano drug delivery systems for the delivery of Nrf2 modulators to the brain, are provided. The literature reviewed shows that the formulations discussed can provide various improvements to the bioavailability and pharmacokinetics of natural Nrf2 modulators. This has been demonstrated in animal models and clinical studies, thereby increasing the potential for the translation of natural Nrf2 modulators into clinical practice.

## 1. Introduction

The research and development of many pharmaceutical products currently represent particularly exciting scientific and technological fields and have had a tremendous impact on modern healthcare. Indeed, such studies are expected to make even greater improvements to the quality of human life. During the evolution of drug delivery in the mid-1960s, most research focused on the controlled release of small drug molecules via conventional forms of dosing. Nanotechnology did not appear in the field of drug delivery until much later, partly because of the lack of sophisticated instrumentation and techniques available until the 1990s [1,2,3,4]. Nanoscience and nanotechnology were then particularly fertile fields in terms of both fundamental discoveries and new tools, as these opened up new worlds of exploration. Nanotechnology has the potential to add innovative functionality to many pharmaceutical products and medical devices. Since then, the research has developed further, with patents registered and new technologies being increasingly translated into clinical testing, new products are already reaching the market [5].

The present review summarises the biological, technological, and nanotechnological discoveries from the field of the currently known natural modulators of nuclear factor erythroid 2-related factor 2 (Nrf2) and their actions in preventing the development of oxidative-stress-related diseases. Oxidative stress occurs when the antioxidant defence system is not able to scavenge the elevated levels of reactive oxygen species (ROS) [6,7,8]. ROS are highly reactive oxygen-containing molecules (e.g., ^•^OH, O_2_^•−^, H_2_O_2_) [8]. At low concentrations, ROS are implicated in several cellular functions and play a vital role as regulators of important signalling pathways, while at high concentrations, they may lead to oxidative damage of the major cellular macromolecules (i.e., proteins, lipids, DNA, and carbohydrates) [6,7,8]. Thus, increased and/or aberrant production of ROS may contribute to various pathological conditions (e.g., cardiovascular, neurological, or metabolic) [9]. Pathologies where ROS have been recognised as contributing factors include cardiovascular disease, diabetes, rheumatoid arthritis, neurodegenerative disorders, and cancer [6,7,8,10,11].

The application of nanotechnology to medicine employs materials that are designed to interact with the body at cellular and subcellular (i.e., molecular) scales with a high degree of specificity [4]. These can then be translated into targeted and tissue-specific clinical applications that are designed to achieve maximal therapeutic efficacies with minimal side effects [12]. Targeting of specific signalling pathways, such as the Nrf2/Kelch ECH associating protein 1 (Keap1) signalling pathway, can provide innovative approaches to tackle oxidative stress, and consequently, its related diseases, which include cardiovascular and pulmonary diseases, diabetes, cancers, and neurodegenerative disorders. There has also been great interest in chemoprevention, especially using natural polyphenolic compounds, which might be the key to preventing the trigger of such diseases. Nrf2 coordinates both the physiological and stress-induced activation of a great number of antioxidant enzymes [13].

Many phytochemicals have long been known to have antioxidant activities, such as curcumin, resveratrol, quercetin, genistein etc. Indeed, the use of curcumin-containing turmeric powder has played a part in traditional medicine in China and India for over 2000 years [14]. Resveratrol has been found in dietary sources such as grapes, blueberries, peanuts, and red wine, and it was first isolated in 1940; since then, it has been extensively studied as an important part of a healthy diet [15]. Many polyphenolic compounds have only recently begun to receive attention following the recognition of their modulatory actions on the Nrf2 pathway [16]. Reports in the literature on the pharmacology of natural Nrf2 modulators are plentiful; however, there is a distinct difference between the study of their delivery and their actions on the Nrf2 pathway. For example, there are currently over 2500 publications on PubMed in the search for “curcumin drug delivery”, whereas a search of the same database for the term “curcumin Nrf2” only produces ~400 results. Among the latter, there is a distinct lack of in vivo studies, as the activities of Nrf2 modulators on their actual targets are usually evaluated in in vitro cell cultures. Furthermore, due to the current status of polyphenolic Nrf2-modulating phytochemicals as potential dietary supplements, their oral delivery is likely to be more relevant for disease prevention rather than disease treatment.

Oral administration has been the most common manner of drug delivery for thousands of years, with the general advantages including safety, good tolerance and compliance, low treatment costs, and convenience to the patients [1,2]. Despite its outstanding advantages, traditionally oral administered drugs are faced with the daunting challenges of poor and highly variable bioavailability. This can be frequently caused by their inherent instabilities and low solubilities under the variable conditions of the gastrointestinal (GI) tract, combined with poor permeabilities through GI barriers, and in some cases, extensive pre-systemic metabolism in the GI tract and liver (e.g., via cytochrome P-450) and rapid clearance. Among many formulations for oral delivery, nanoparticulate drug delivery systems (nanoDDSs) have attracted great attention, as they can protect drugs from premature degradation and improve their interactions with the physiological environment, enhancing solubility and drug absorption, and increasing intracellular penetration [17,18].

Research into cancer nanotechnology has recently defined passive targeting approaches, such as the delivery of nanoparticles in angiogenesis as a compensatory mechanism of diffusion in cancers [19]. The enhanced permeability and retention (EPR) effect has been exploited in numerous studies covering the design of nanocarriers for the treatment of cancer [20]. Passive targeting depends upon the properties of the nanoparticles, such as their size, shape, surface properties, and mechanical stiffness [21]. Passive targeting also involves the use of other innate characteristics of nanoparticles that can affect tumour targeting. The ability of nanoDDSs to passively target tumours has been shown many times in in vivo animal models through the use of different imaging techniques [22,23]. Unfortunately, passive targeting based on the EPR effect has shown heterogenous and questionable efficacy in tumour targeting. A recent meta-analysis study revealed that less than 1% of the nanocarriers accumulate in xenografted tumours [24]. To surpass this challenge and improve tumour targeting, nanocarrier surfaces can be modified with ligands, enabling active targeting [20]. The wide selection of publications point to the great potential and future perspectives of nanotechnology in cancer treatment.

Despite the clear lack of data outlining the effects of the plasma concentrations of natural Nrf2 modulators on clinical outcomes, their increased efficacies based on improved pharmacokinetic profiles can be expected. We note that natural Nrf2 modulators have been frequently studied by different groups using different methods and under different experimental conditions, which greatly hinders the relevant comparability of the results obtained. Therefore, this review is constructed with a focus on a more extensively reported research area focusing on improvements to their oral delivery. The aim here is to provide the reader with a basic mechanistic background of Nrf2 modulators and the limitations of their use. The most commonly studied formulations that have improved their pharmacokinetic parameters are discussed, along with selected examples that have been evaluated in in vivo models. This paper evaluates the current state-of-the-art technology of natural Nrf2 modulators and their delivery.

## 2. Cellular Mechanisms and Modulation of Nrf2 Response to Oxidative Stress

The constant production of reactive oxygen (ROS) and nitrogen (RNS) species in the human body due to endogenous (e.g., metabolism) and exogenous factors can lead to oxidative stress if the cellular antioxidant defence system cannot maintain redox homeostasis to thus neutralize the elevated ROS and RNS levels [7,25]. One of the key players to combat oxidative stress is Nrf2, which was identified in humans in 1994 [26]. Nrf2 is a 605-amino-acid protein that belongs to the cap’n’collar subfamily of basic-region leucine-zipper-type transcription factors [27]. Together with cysteine-rich Keap1, a repressor protein that binds to Nrf2 and facilitates its degradation via the ubiquitin-proteasome pathway [28], these constitute the Nrf2/Keap1 signalling pathway, which is a key player in the regulation of cytoprotective responses to oxidative and electrophilic stresses. When oxidative stress occurs, the modification of the reactive cysteine residues of Keap1 triggers the release of Nrf2. Nrf2 then undergoes translocation into the nucleus, binding to the antioxidant response element (along with the small musculoaponeurotic fibrosarcoma[Maf proteins) to facilitate the expression of key endogenous antioxidant enzymes (Figure 1) [29]; i.e., NAD(P)H quinone oxidoreductase 1, superoxide dismutase, catalase, heme oxygenase 1, glutamate–cysteine ligase, and glutathione S transferases [28,30,31]. Furthermore, the target genes of Nrf2 are also associated with glutathione synthesis, xenobiotic metabolism, and drug transport [30].

Regulation of Nrf2 activity is carried out through several mechanisms that involve both Keap1-dependent (canonical pathway) and independent (non-canonical) regulation of Nrf2. These mechanisms can affect the stability or degradation of Nrf2, and include post-translational modifications, where Nrf2 undergoes phosphorylation, acetylation and interaction with distinct partners, and Nrf2 regulation at the transcriptional level and through other regulatory mechanisms [32,33,34].

As the modulation of the Nrf2/Keap1 signalling pathway is one of the main defence mechanisms against oxidative and electrophilic stress, this is of great interest for novel drug development due to the potential targeting of many oxidative-stress-related diseases [30]. The connection between oxidative stress and the pathogenesis of many neurodegenerative diseases (e.g., Alzheimer’s, Parkinson’s and Huntington’s diseases, multiple sclerosis, Friedrich’s ataxia, and stroke) is inevitable [35]. Therefore, it has been suggested that the activation of Nrf2 can attenuate oxidative stress and many other processes that are known to trigger neurodegeneration (e.g., mitochondrial dysfunction and neuroinflammation). The delivery of Nrf2 activators to the central nervous system is thus of great interest for future research.

The Nrf2/Keap1 signalling pathway is also known to be closely related to cancers [29,31,32,36,37,38]. In normal healthy cells, Nrf2 assists in the maintenance of cellular redox homeostasis via the elimination of ROS and chemical or physical carcinogens, while in several types of cancer, the overexpression of Nrf2 enables the cancer cells to adapt to a hostile microenvironment, whereby they can become resistant to elevated endogenous ROS levels and to chemotherapeutic agents and radiotherapy [29,36,37,39]. One approach to prevent cancer development is to induce cytoprotective enzymes that can remove toxic reactive species that arise as a result of carcinogens [40]. Thus, in non-malignant cells, the protective role of Nrf2 is established via its activation and the consequent increase in antioxidant defence; however, in the process of tumour development, cancer cells can ‘use’ Nrf2 activation to overcome a hostile microenvironment and to diminish the effectiveness of chemotherapy and radiotherapy [36,40]. In the latter case, the use of Nrf2 inhibitors to sensitise cancer cells to established therapies would be preferred.

## 3. Current Status and Limitations of Natural Nrf2 Modulators

The beneficial effects that can result from the regulation of Nrf2 represent an important strategy for the Nrf2-based prevention of oxidative-stress-related diseases [32]. Chemoprevention might be achieved by natural products via modulation, i.e., activation of the Nrf2 signalling pathway. The majority of Nrf2 activators are electrophiles that can react with the cysteine residues of the thiol-rich Keap1 protein [41]. Many natural electrophilic Nrf2 inducers have been identified, and these can be classified into several distinct classes according to their chemical natures [40]. The class of Michael acceptors includes flavonoids (e.g., flavanole, genistein, kaempferol, luteolin, quercetin), chalcones, coumarins, terpenoids, curcuminoids (e.g., curcumin), and cinnamic acid derivatives. Resveratrol is one of the most studied polyphenols, epigallocatechin gallate is a representative of the oxidisable phenols and quinones, and sulforaphane is the most important Nrf2 inducer among the isothiocyanates.

The natural Nrf2 activators resveratrol, curcumin, sulphoraphane, and quercetin (Table 1) are currently involved in several clinical trials for various therapeutic indications [13,41]. The proposed general mechanism of action of natural polyphenolic antioxidants relies on the alkylation of Keap1, which occurs after the polyphenol is oxidised to the corresponding ortho-quinone or para-quinone [42]. Keap1 alkylation releases Nrf2 from the complex, thus avoiding its ubiquitinylation and degradation. The subsequent translocation of Nrf2 to the nucleus then triggers the expression of the target genes to establish an effective antioxidant defence.

On the other hand, in the case of some types of cancers, Nrf2 inhibitors can be used to sensitize cancer cells to established anti-cancer therapies. The most common natural Nrf2 inhibitors are quassinoid brusatol from Brucea javanica, the flavonoids luteolin (Table 1), apigenin and chrysin, the coffee-derived alkaloid trigonelline, and all-trans retinoic acid [29].

Most of the natural Nrf2 modulators face several challenges and limitations in terms of their effectiveness, e.g., their bioavailability, the concentrations that can be realised in vivo, their location in target cells, and their cellular metabolism, among others [42]. Their low biological concentration can be especially problematic, as most Nrf2 activators are naturally electrophilic and are rapidly metabolised [13]. Moreover, most common natural Nrf2 modulators (e.g., sulforaphane, curcumin, triterpenoids) activate Nrf2 by increasing oxidative stress, thus raising the oxidative load of already delicate cells [43]. One of the challenges is also to reduce the risk of their non-specific effects, as many natural Nrf2 modulators can act on other signalling pathways and alter related biological processes [13]. This is especially important for the delivery of Nrf2 activators to the brain for the treatment of central nervous system disorders or for the potential use of Nrf2 inhibitors, as their systemic side effects might be problematic if they do not reach the specific target cell or organ.

Currently, curcumin and resveratrol are among the most commonly studied natural modulators of cellular mechanisms involved in the pathologies of cancers and other oxidative stress-related diseases, including the Nrf2 pathway [29,52]. Additionally, there is currently the promise of the ‘rediscovery’ of many other natural Nrf2 modulators, based on their potential for the modulation of stress-related diseases. The positive effects of natural Nrf2 activators on cellular redox homeostasis might be expected when they are ingested with food or as a dietary supplement. However, to increase and further exploit these modulatory effects in the prevention of oxidative stress-related diseases, higher concentrations of these compounds need to be delivered to the body. Thus, this illustrates the need for the development of drug-delivery strategies for the optimisation of Nrf2 modulators in a clinical setting.

As indicated, oral administration has multiple benefits of convenience, low cost, safety, lack of need for trained medical personnel, and an absence of patient discomfort. Thus, this is the most acceptable administration route for patients [53], which also makes it an obvious choice for the implementation of Nrf2 modulation for the prevention of relevant diseases. This is especially the case for long-term use, as it is expected that selected Nrf2 activators for cancer prevention would require regular administration over extended periods of time. Sulforaphane [54] and dimethyl fumarate [55] are the only widely studied Nrf2 activators that are characterised by high solubility and membrane permeability. Unfortunately, most other natural Nrf2 modulators with potential for the prevention or treatment of cancers, and other oxidative stress-related diseases (e.g., polyphenols in Table 1), are plagued by limited oral bioavailability [52].

Although polyphenols usually exhibit high permeability in terms of crossing biological membranes, they are also characterised by low aqueous solubility. This characteristic is particularly relevant for natural polyphenolic Nrf2 modulators such as resveratrol [50], curcumin [56], luteolin [48], and others (Table 1). As such, many known natural Nrf2 modulators are classified as part of class II of the Biopharmaceutical Classification System (BCS) as they have low oral bioavailability due to limited aqueous solubility [52,57,58,59]. For example, oral curcumin doses that yield detectable plasma concentrations in humans are in the range of several grams, which is not acceptable for patients [60,61].

In addition to low aqueous solubility, the oral bioavailability of polyphenols is further limited by their extensive pre-systemic metabolism in the liver and their rapid elimination, which both decrease the plasma concentrations [62]. For example, resveratrol undergoes glucuronidation and sulphate conjugation in the liver, and >98% of it is metabolised pre-systemically after the administration of a moderate dose of 25 mg/70 kg body weight [62,63]. The hepatic metabolism of luteolin results in its monoglucuronide form [64], and curcumin is rapidly metabolised to hexahydrocurcumin and tetrahydrocurcumin before undergoing conjugations to glucuronide and sulphate forms [60,65]. Furthermore, an additional step in metabolism is introduced in the intestine, where polyphenols such as resveratrol [66] and curcumin [67] have also been shown to be metabolised by the gut microbiota. In addition, after systemic absorption, the elimination of natural polyphenols is relatively rapid and results in a high clearance, thereby even further limiting systemic exposure [53]. The extensively studied polyphenolic example of resveratrol shows a rapid decrease in plasma concentration [68]. These factors all contribute to the limited oral bioavailability of these natural Nrf2 modulators (Figure 2).

Polyphenols are known to have photochemical instability. This can be exemplified by the light-induced isomerisation of resveratrol from its bioactive isoform trans-resveratrol to the biologically less active form of cis-resveratrol [52,69]. Therefore, pharmaceutical approaches might even be beneficial in order to increase the stability of Nrf2 modulators by offering protection from environmental factors that have the potential to induce instability and thus to decrease the biological activity during storage [70].

Increasing the oral bioavailability of poorly soluble Nrf2 modulators through formulation strategies is imperative. The currently promising formulation strategies to increase the oral bioavailability of Nrf2 modulators are solid dispersions [71], self-microemulsifying DDSs (SMEDDSs) [72,73], solubilising approaches through the use of micelles [74], and nanoformulation approaches, that are based on nanonisation of drug particles [75,76] or the use of nanocarriers [77,78]. The above-mentioned formulation approaches for the oral delivery of Nrf2 modulators are presented in a critical overview with a focus on nano-delivery systems as a growing research topic (Figure 2).

## 4. Formulation Approaches to Improve Oral Bioavailability of Natural Nrf2 Modulators

This section provides a review of the most commonly studied formulation approaches for the mitigation of poor oral bioavailability in natural Nrf2 modulators. Due to the extensive literature on this topic, the examples included are selected as representative of the current trends for the formulation of these compounds. These studies include the optimisation of the delivery of Nrf2 modulators as shown through comparisons of the formulated and non-formulated compounds in pharmacokinetic studies in in vivo models and/or on the basis of improved in vitro solubility and dissolution. The chosen and reviewed in vivo studies include only a few clinical pharmacokinetic studies in humans, whereas the majority of pharmacokinetic evaluations were performed in rodent (mice or rat) models with the exception of two studies in a rabbit model. These included enteric modes of administration (described either simply as oral administration or oral gavage). At least two groups were compared—animals administered with the non-formulated or native compound and animals administered with the evaluated formulation containing the respective compound. The doses of the formulated and non-formulated compounds were the same in all studies unless otherwise stated—there, the improvement of bioavailability was normalised to a dose of administered phytochemical. Across the different in vivo studies, the animals were administered different doses of polyphenolic phytochemicals, ranging from 20 mg/kg to 250 mg/kg. The blood samples were withdrawn from different arteries and veins, and concentrations of the studied compounds were determined in the plasma chromatographically.

### 4.1. Solid Dispersions

Solid dispersions are widely applied as a strategy to increase drug solubility in oral formulations. Indeed, they have been gaining more and more interest due to their great potential for the improvement of the bioavailability of BCS class II drugs, including poorly soluble natural Nrf2 modulators. This strategy is promising due to the relative ease of integration of a solid dispersion into a solid oral dosage form appropriate for long-term patient-friendly administration. The improvement of oral bioavailability with solid dispersions is based on the dispersion of the drug in an inert hydrophilic carrier (usually a polymer, surfactant or combination of these) that increases its solubility and dissolution rate [79,80]. These improvements are due to the larger specific surface area of the drug and its improved wettability due to the presence of the hydrophilic matrix. Furthermore, the drug can be present in a solid dispersion in an amorphous state, which is characterised by higher kinetic solubility and a higher dissolution rate than in the crystalline state. Thus, the drug can be dissolved to result in a supersaturated solution which results in its accelerated diffusion across the epithelium [81,82,83,84]. Another advantage of solid dispersions is the possibility for their fabrication using novel techniques such as the supercritical antisolvent process [85] or supercritical-assisted atomisation. These have recently been reported as methods for the preparation of solid dispersions with natural Nrf2 modulators [86].

A solid dispersion containing a mannitol and D-α-tocopheryl polyethene glycol 1000 succinate (TPGS) matrix with amorphous curcumin was studied in a pharmacokinetic study in a rat model. This was an impressive solid dispersion for the delivery of curcumin and exhibited a 65-fold increase in area under the curve of the plasma concentration profile (AUC) and an 86-fold increase in the maximal plasma concentration (c_max_) compared to the oral administration of non-formulated curcumin as an aqueous dispersion [71]. Several other curcumin solid dispersions based on other polymers (e.g., hydroxypropyl methylcellulose, arabinogalactan) have also increased the drug’s oral bioavailability in in vivo animal models, with the consequent improvements seen by greater than 17-fold and 60-fold increases in the AUC and c_max_, respectively [87,88,89]. Resveratrol-containing polymer-based solid dispersions with different polymers (e.g., Eudragits, hydroxypropyl methylcellulose, Soluplus, poloxamer 407) [90,91] and with inorganic mesoporous silica microparticles [92] have also been evaluated, with improvements in their oral bioavailability shown in animal models (Table 2).

An impressive improvement of oral bioavailability was shown for the Nrf2 modulator quercetin, Quercetin Phytosome^®^, as also demonstrated in a clinical pharmacokinetics study [94]. The Quercetin Phytosome formulation was shown to increase its bioavailability by >20-fold. Quercetin Phytosome is a supramolecular assembly that is formed by interactions between lecithin and quercetin. Upon oral administration, quercetin is solubilised by the lecithin micelles that form in the aqueous medium in the GI tract in a manner reminiscent of SMEDDSs. However, in its solid state, Quercetin Phytosome is structurally more comparable to a solid dispersion than a SMEDDS [100].

Solid dispersions thus represent a promising formulation approach for the improvement of the oral bioavailability of natural Nrf2 modulators (Table 2). Their biggest advantage are the state-of-the-art technologies for their production, which are already widely used on an industrial scale [82]. Therefore, the continued presence of solid dispersions in the oral delivery of poorly soluble Nrf2 modulators can be expected.

### 4.2. Self-Microemulsifying Drug Delivery Systems

Self-microemulsifying drug delivery systems (SMEDDSs) are another promising strategy for the improvement of the oral bioavailability of poorly soluble Nrf2 modulators, with the possibility for their integration into solid oral dosage forms [101]. Similar to solid dispersions, SMEDDS can increase the solubility and dissolution rate of a drug. The SMEDDS-forming excipients solubilise the drug by the spontaneous formation of mixed micelles composed of the SMEDDS excipients, bile salts, endogeneous bile lipids, and the GI fluid [102,103]. However, unlike solid dispersions, SMEDDSs contribute an additional mechanism to increase the bioavailability: the circumvention of pre-systemic hepatic metabolism by the promotion of lymphatic uptake, increasing uptake via the inclusion of the drug in the triacylglycerol-based lipoproteins known as chylomicrons. The latter are capable of lymphatic uptake [102,104,105]. Polyphenolic Nrf2 modulators are excellent candidates for delivery by SMEDDSs due to their low solubilities and extensive first-pass metabolism.

There have been several reports on the use of SMEDDSs for curcumin, resveratrol, and luteolin delivery. Tang et al. reported a dramatic increase in resveratrol solubility in water (>1000-fold) when incorporated into a SMEDDS of isopropyl myristate, polyethene glycol (PEG)400, and Cremophor RH40; unfortunately, the improved oral bioavailability of this formulation was not evaluated in vivo [73]. Jaisamut et al. reported on the most successful increase in the oral bioavailability of curcumin among SMEDDS formulations in an in vivo animal model, as a 43.7-fold relative bioavailability and a 30.7-fold increase in c_max_ (to 5.84 μg/mL), compared to non-formulated curcumin [95]. Most other SMEDDS formulations tested in in vivo animal models have shown lower increases in the oral bioavailability of polyphenols, although these have often still been by severalfold and are thus significant and promising [97,98,99].

The highlighted reports show promise for the use of SMEDDSs for improvements in the solubilities of Nrf2 modulators plagued by low oral bioavailability. However, despite some dramatic increases in dissolution rates and solubilities, in vivo studies on animals have not shown the same factor of increase in oral bioavailability in comparisons of the non-formulated and SMEDDS-formulated Nrf2 modulators. Moreover, as SMEDDSs contain relatively large quantities of surfactants, such as Tweens, poloxamers, polyoxylglycerides, and polyethoxylated castor oil, their potential toxicity and irritability in the GI tract must be considered in terms of the long-term application that might be required for disease prevention [95].

### 4.3. Nanoformulations for Oral Delivery of Nrf2 Modulators

Formulations containing nanoDDSs are interesting choices for the oral administration of poorly soluble Nrf2 modulators with low oral bioavailability. With the great potential of nanotechnology, the use of nanoDDSs for the prevention and treatment of oxidative stress-related diseases is expanding. NanoDDSs can provide improved solubility of poorly soluble drugs [106]. For improved oral bioavailability, Nrf2 modulators can be formulated as carrier-free nanoparticles (i.e., pure drug nanocrystals) or can be encapsulated into different types of nanoparticulate carriers. As there have been many studies on nanoDDSs for Nrf2 modulator administration, here we provide an overview of selected representative reports of commonly studied nanoDDSs that indicate the potential of these systems for the future.

#### 4.3.1. Nanocrystals

Structurally, the simplest example of a nanoformulation is nanocrystals of the particular drug, which represent a carrier-free approach to increase oral bioavailability. The increase in the specific surface area of the drug nanocrystals leads to an increased dissolution rate. This allows for the formation of a supersaturated solution of the drug in the GI lumen, thereby increasing the absorption rate [107,108,109]. Nanocrystals have already been shown to increase dissolution rates for curcumin [75] and resveratrol [76,110] compared to crude compounds. They have also provided improved the oral bioavailability of these compounds in animal models, as seen by the severalfold increases in AUC and c_max_ compared to administration of larger free drug particles [110,111]. A negative aspect of nanocrystals presents itself in their high surface free energy, which increases the potential for agglomeration, and thus necessitates the use of stabilisers. To this end, small amounts of surfactants are used to prevent this agglomeration, which results in physical instability during storage, thus counterbalancing the positive effects on the dissolution rate and oral bioavailability [110,112]. Potential stabilisers for nanocrystals of polyphenolic Nrf2 modulators reported in the literature include polyvinyl alcohol [113,114], sodium dodecyl sulphate [114], d-α-tocopherol polyethene glycol 1000 succinate [76,113,114], lecithin [110], pluronics [110], polyvinilpyrrolidone [113,114], sodium carboxymethylcellulose [113], hydroxypropyl methylcellulose, Tweens, and poloxamers [76].

#### 4.3.2. Carrier-Based Nano-Delivery Systems

Carrier-based nanoDDSs have also been frequently studied for the improvement of the solubility and oral bioavailability of natural Nrf2 modulators. These have included nanoDDSs such as polymeric nanoparticles [115], solid lipid nanoparticles [116], micelles [117], nanoemulsions [118], liposomes [119,120], and dendrimers [121]. The improvement of the Nrf2 modulator oral bioavailability by encapsulation in nanoDDSs is based on the following mechanisms: (i) protection of the drug from enzymatic degradation in the GI tract (e.g., curcumin, which shows such instability); (ii) increased solubility and dissolution rate of the drug; (iii) adhesion-mediated increased gastric or intestinal residence time; and (iv) absorption of the entire nanocarriers. In the last case, the nanocarrier can protect the drug from first-pass metabolism and rapid elimination during systemic circulation [115,122]. The transepithelial uptake of nanocarriers is likely to be relatively low. This arises from the very limited transepithelial transport of entire nanoparticles, as the barriers of mucus and the intestinal epithelium impede their intestinal uptake [123]. Literature reports on nanoparticle-mediated increases in the oral bioavailabilities of natural Nrf2 modulators lack data on the presence of nanoparticles or their constituting excipients in the plasma, thereby preventing further conclusions regarding the contribution of transepithelial absorption of intact nanoparticles. Furthermore, as most natural Nrf2 modulators with low oral bioavailability have high membrane permeability, the incorporation of BCS class II Nrf2 modulators into nanoparticles with the expectation of significant transepithelial absorption might appear an irrational strategy. Consequently, significant nanocarrier-mediated protection of natural Nrf2 modulators and the subsequent impediment of their rapid biotransformation and elimination are not to be expected in the current state-of-the-art oral nanosized delivery systems [124].

Nanocarriers have been studied extensively above all for oral administration of Nrf2 modulators, and they have shown some promising results in pharmacokinetic studies in animal models [125,126,127,128]. The following section outlines the most frequently studied applications of nanocarriers as nanoDDSs for the oral administration of Nrf2 modulators (Table 3).

#### 4.3.3. Polymeric Nanoparticles

Polymeric nanoparticles have been widely studied for many applications, including increased oral bioavailability [148,149,150]. Due to the availability of many natural and synthetic polymers with different physicochemical properties, polymeric nanoparticles are one of the most diverse groups of nanocarriers. The different polymers offer the possibility to prepare nanoparticles with ‘tunable’ release properties, such as pH-dependent release or sustained release due to biodegradation [84]. It is, therefore, no surprise that polymeric nanoparticles are the most commonly studied nanoDDS for the improvement of the delivery of natural Nrf2 modulators. The most widely studied polymers in nanoparticles intended for oral administration are poly(lactic-*co*-glycolic acid) (PLGA), polycaprolactone (PCL), chitosan and its derivatives, and polymethacrylates (i.e., Eudragits) [151,152].

Low aqueous solubility as a limiting factor in the dissolution profiles of non-formulated natural Nrf2 modulators can be combated by encapsulation in polymeric nanoparticles [148,149]. High loading of the drug into the polymeric nanoparticles must be achieved in order to fully exploit their potential to increase the drug solubility. In addition, the factor of solubility increase might not be paralleled by the increase of oral bioavailability. For example, the incorporation of curcumin into PLGA-based polymeric nanoparticles provided a 640-fold increase of in vitro solubility, while oral administration of the same nanoformulation in a rodent model resulted in only a 5.6-fold increase in bioavailability when compared to the non-formulated drug [115]. Poor aqueous solubility of natural Nrf2 modulators results not only in low amounts of the total released drug but also in slow dissolution rates. Faster dissolution rates can increase the concentration gradient of BCS class II compounds and thus promote their more efficient and rapid diffusion across the intestinal wall [153]. As such, superior dissolution profiles of drugs in nanoparticle formulations are indicative of their potential to increase oral bioavailability [148]. This feature of polymeric nanoparticles for oral administration is no different for poorly soluble natural Nrf2 modulators. Many studies have reported concurrent in vitro dissolution and in vivo pharmacokinetics studies on polymeric nanoparticles for the oral delivery of different natural Nrf2 modulators. As expected, increased dissolution rates and cumulative release in vitro result in higher oral bioavailabilities in in vivo animal models, where the increased bioavailability of natural Nrf2-modulating polyphenols has usually been in the range of 2-fold to 10-fold [125,133,154,155,156].

The most promising nanoparticle-mediated improvement of a pharmacokinetics profile was achieved by a formulation of curcumin in PEG-PLGA nanoparticles for oral administration. This formulation provided a 55.4-fold increase in curcumin AUC after oral administration, compared to non-formulated curcumin administered as a suspension. An increase in c_max_ indicated an improvement in curcumin absorption [125]. PLGA nanoparticles are an excellent carrier that can improve the solubility of hydrophobic drugs such as natural Nrf2 modulators [115]. PLGA nanoparticles have provided the increased oral bioavailability of some natural poorly soluble Nrf2 modulators in in vivo animal models, and they are especially interesting with regard to their sustained release of lipophilic compounds (e.g., natural polyphenols). PLGA nanoparticles for the oral delivery of curcumin have provided increases in oral bioavailability ranging from 5-fold to 55-fold [115,125,129]. PLGA NPs can also be employed for the sustained release of polyphenolic Nrf2 modulators. This can result in an increased peak time (t_max_) and a prolonged duration of the drug in the systemic circulation, which was shown in animal models for the oral administration of curcumin-loaded PLGA nanoparticles [115,129]. For example, in one study, the increased t_max_ was accompanied by a curcumin plasma concentration profile of over 48 h (compared to unformulated curcumin, at 6 h), thus indicating the prolonged release of curcumin from the PLGA nanoparticles [129].

In addition to PLGA nanoparticles, chitosan-derivative-based nanoparticles have frequently been investigated as potential oral delivery systems for natural Nrf2 modulators. Carboxymethyl-chitosan-based nanoparticles containing resveratrol have shown increased dissolution rates and increased AUC in pharmacokinetics studies after oral administration in rats [133], and trimethyl-chitosan-based nanoparticles containing resveratrol have shown an increased uptake of resveratrol into Caco-2 cells in in vitro cell monolayer models [157]. Polyphenols cross the intestinal epithelium by passive diffusion both transcellularly and paracellularly. Chitosan is believed to increase tight junction permeability of the intestinal epithelium [158,159], which means that chitosan-based nanoparticles can potentially stimulate the absorption of polyphenols based on the paracellular pathway as well. The possible mechanisms behind the increased absorption of polyphenolic Nrf2 modulators after oral delivery with chitosan nanoparticles are therefore multiple: increased transcellular and paracellular absorption, as well as the prolonged residence time of nanoparticles at the intestinal epithelium, and the subsequent increased local resveratrol concentration gradient due to the mucoadhesive properties of chitosan [160,161].

Oral delivery using polymeric nanoparticles has also shown the improved efficacy of curcumin in an in vivo mouse model. Tripathi et al. showed the improved oral bioavailability and improved behavioural tests and anti-stress activity in mice after oral administration of curcumin-loaded poly (amidoamine) dendrimer-palmitic acid core-shell nanoparticles [128]. However, due to the multiplicity of the possible mechanisms of action of curcumin, such tests alone lack any data on improvement in the activation of Nrf2.

#### 4.3.4. Solid Lipid Nanoparticles

Solid lipid nanoparticles (SLNs) are nanoparticles that are composed of lipids that are present in their solid state at room temperature [162]. The main advantage of SLNs is their composition, which is based on biocompatible lipids and surfactants that are generally recognised as safe and thus pose a minimal threat of inducing biotoxicity [163]. The modified (e.g., extended) release of Nrf2 modulators into the systemic circulation can be achieved using SLNs [141]. Due to the hydrophobicity of their interior, they provide the high encapsulation efficiency of lipophilic compounds [164,165], making SLNs an excellent choice as delivery systems for polyphenolic natural Nrf2 modulators.

It is generally accepted that SLN-mediated improvements in the oral bioavailability of poorly soluble drugs are based on the following mechanisms: increased solubility; protection from the harsh conditions of the GI tract; and protection from first-pass metabolism and rapid systemic elimination [166]. It is believed that the absorption of poorly soluble drugs can be improved by SLN-mediated increased residence time in the intestine. This has been shown by the improved oral bioavailability of Nrf2 modulators [139]; however, the degree of correlation between the two phenomena is unclear (Table 3).

The main advantage of SLNs over other nanocarriers is their mediation of lymphatic uptake, which contributes to the evasion of their hepatic metabolism. This is especially relevant for drugs that show an extensive first-pass metabolism, as for most natural Nrf2 modulators. Increased lymphatic uptake of curcumin mediated by SLNs has been reported and resulted in improved systemic curcumin exposure [77,116]. SLNs can cross the epithelial barrier via two transcellular routes that enable lymphatic uptake: the pathway through enterocytes and the pathway through M-cells of Peyer’s patches. During the former route, the poorly soluble Nrf2 modulator can be solubilised in the chylomicrons that they formed in enterocytes. The drug is then delivered by the chylomicrons into the lymphatic circulation via the mesenteric node [77,116,167]. The second transcellular route through M-cells results in direct lymphatic uptake of SLNs [163,168,169]. Further increased lymphatic uptake can be achieved by the addition of chitosan or its derivatives into SLN structures. Furthermore, coating SLNs with chitosan or its derivatives has also been shown to mitigate the burst release of polyphenols into the acidic environment, thus protecting them from extensive premature release into the gastric fluid [77,116,126].

There have been several reports of orally administered SLNs that have improved the pharmacokinetic profiles of natural Nrf2 modulators. Kakkar et al. (2010, 2011) achieved the highest SLN-mediated increases in curcumin AUC and cmax in a rat model (39.1-fold, 48.9-fold, respectively) [135,136]. Similarly to other oral formulation approaches, most other SLNs have been reported to provide AUC increases of up to 10-fold in pharmacokinetic studies in rodent models [126,141,170]. For example, stearic acid and glycerol monostearate-based SLNs exhibited 8.0-fold and 3.6-fold increases in AUCs of resveratrol [141] and luteolin [170], respectively. Regardless of the administration route, SLNs can increase the systemic exposure of natural Nrf2 modulators by conferring protection from metabolism and elimination, as discussed above. SLN-mediated protection from rapid elimination has been shown by increased polyphenol half-lives in several in vivo studies [137,139,166]. Additionally, the mitigation of rapid elimination by encapsulation in SLNs has been further supported based on the increased accumulation of resveratrol in the liver and kidney of rats, where SLNs are believed to protect natural polyphenols from biotransformation [140]. SLNs also have great potential for the intracellular delivery of Nrf2 modulators, as they have been shown to be taken up by a variety of different cells, such as keratinocytes [16], A549 cancer cells [171,172], macrophages [173], Caco-2 cells [174], and cervical HeLa cells [172].

#### 4.3.5. Micelles

Micelles (especially polymeric and mixed micelles) can improve the oral bioavailability of poorly soluble natural Nrf2 modulators by increasing solubility via their solubilisation. Polymeric micelles are formed from amphiphilic molecules with copolymeric structures that contain hydrophilic and hydrophobic blocks. Mixed micelles, on the other hand, contain an amphiphilic copolymer and at least one additional component from the class of either low molecular weight surfactants or amphiphilic copolymers. The advantage of both polymeric and mixed micelles is their lower critical micellar concentration than is typical of conventional surfactant-based micelles, which provides them with greater stability upon administration and dilution [145,175]. Polymeric micelles are an appropriate strategy for the oral delivery of natural Nrf2 modulators, as they show stability in biological media and can therefore confer protection from the environment of the GI tract. Additional mechanisms for the improvement of oral bioavailability by polymeric micelles include increased intestinal retention time by mucoadhesion and the inhibition of efflux pumps (e.g., P-glycoprotein) in the intestinal wall [176,177].

In in vivo rodent models, both polymeric and mixed micelles have been shown to achieve up to 10-fold increases in the oral bioavailability of commonly studied poorly soluble natural Nrf2 modulators, such as for curcumin [127], resveratrol [145], quercetin [146,147], and genistein [144] (Table 3). In contrast to the lack of clinical pharmacokinetics data on the nanoparticle-mediated delivery of Nrf2 modulators, there are also reports on human pharmacokinetics data that have indicated their improved oral delivery by micelles [74,143]. The most significant success in oral bioavailability improvement for curcumin in humans was reported for Tween 80 micelles, with 185-fold AUC and 453-fold c_max_ increases (for a c_max_ of 3228 ng/mL, across both males and females) [143]. This confirmation of the promise of micelles is extremely relevant nowadays when many nanoformulations show low efficiency for their translation from the pre-clinical to the clinical settings [177].

## 5. Nanotechnological Approaches to Improve the Delivery of Natural Nrf2 Modulators to the Brain

Much of this review has focused on the disadvantages of natural Nrf2 modulators and the potential solutions for their oral administration. However, it is also important to acknowledge the potential of nanoDDSs for parenteral administration, which is currently the main focus in the field of nanomedicines. A significant obstacle to the successful delivery of high concentrations of intravenously administered natural Nrf2 modulators is their rapid elimination. Nanocarriers can confer protection from this rapid elimination, which results in increased half-lives of these natural Nrf2 modulators. Decreased drug clearance by nanoparticle encapsulation can be achieved by protection from metabolic enzymes in the liver or reduction in renal clearance (particle size cut-off, <15 nm) [122]. However, it must be noted that the elimination of nanocarriers from the blood by the reticuloendothelial system can reduce the circulation time of the encapsulated drug. To this end, nanoparticles must be carefully designed to evade rapid elimination, where the opsonisation of nanoparticles and their sequestration in macrophages is heavily dependent on their size, surface charge, and hydrophobicity [178,179].

Another feature of nanoDDSs is the possibility to passively target diseased tissues. Logically, passive targeting on the basis of the enhanced permeability and retention effects can only be exploited for the delivery of inhibitors of Nrf2. Such inhibitors are indeed being studied as cancer therapies due to their sensitisation of cancer cells to chemotherapy, as indicated above. In contrast, Nrf2 activators are more appropriate for the treatment of other oxidative stress-related diseases. Therefore, Nrf2 inhibitors can be delivered by nanocarriers to cancer cells based on the enhanced permeability of the diseased tissue for nanoparticles [180].

Perhaps one of the most interesting modalities of nanocarrier-mediated delivery to target Nrf2 is increased drug bioavailability in the brain. Due to their lipophilic nature, many widely studied Nrf2 modulators have been shown to penetrate the blood–brain barrier (BBB) in animal models, including resveratrol [181], curcumin [182], and genistein [183]. However, the passage of hydrophilic metabolites of these compounds across the BBB is more limited. This is because the lipophilicity of many natural Nrf2 modulators decrease upon glucuronidation and sulphation in the liver (e.g., flavonoids, in particular), and therefore evasion of hepatic metabolism is preferential to increase brain bioavailability [184]. Therefore, even if plasma concentrations of poorly orally bioavailable Nrf2 modulators are increased by using suitable oral formulation approaches or by parenteral administration, additional strategies are needed to deliver these compounds to the brain.

Generally, brain delivery can be achieved by different strategies of overcoming the BBB: (i) diffusion-based transport (either by paracellular or transcellular transcytosis), (ii) transporter-mediated transcytosis, (iii) receptor-mediated transcytosis, (iv) adsorptive-mediated transcytosis, and (v) cell-mediated transcytosis. While diffusion-based and transported-based transcytosis is only possible for individual small molecules with appropriate physicochemical properties, the other pathways can be targeted by nanocarriers in order to improve brain delivery.

Transporter-mediated transcytosis refers to the specific binding of small molecules onto their respective transported on the BBB and their subsequent internalisation into endothelial cells. This mechanism of BBB transport can be targeted to facilitate the penetration of nanocarriers. Examples include surface modification of nanoparticles with glutamate for targeting of large amino acid transporter 1 (LAT1) [185] and conjugation of glucose derivatives to nanoparticles to achieve glucose transporter (GLUT)-mediated BBB permeation [186]. Receptor-mediated transcytosis of entire nanocarriers can be facilitated through modification of their surface by conjugation with appropriate ligands. To this end, several different ligands have been employed, among them lactoferrin [187,188], transferrin [189], and lipoprotein [142,190]. Receptor-mediated transcytosis also presents a potential for increasing brain bioavailability of polyphenolic Nrf2 modulators. For example, polylactic acid-coated mesoporous silica nanoparticles conjugated with a lipoprotein receptor-targeting peptide and containing resveratrol were able to increase this phytochemical’s permeability across an in vitro BBB model by targeting the low-density lipoprotein receptor [191].

Adsorptive-mediated transcytosis or pinocytosis can be exploited by designing nanocarriers with a positively-charged surface that are capable of interactions with the negatively-charged endothelial cell membranes. Electrostatic interactions between nanocarriers enable their higher uptake into the endothelial cells of the BBB [192]; however, non-specific binding is of concern as it could result in the accumulation of positively-charged nanocarriers in non-target tissues. In order to increase penetration of nanocarriers across the BBB, surface modification with positively-charged glycoproteins (e.g., lectin [193]) or peptides can be employed. The conjugation of nanocarriers to cell-penetrating peptides (CPP) has been frequently discussed for the facilitation of adsorptive-mediated transcytosis [194,195,196]. The conjugation of brain-targeting cyclic peptide onto the surface of PLGA nanoparticles was reported to increase the transport of curcumin and an amyloid β generation inhibitor across an in vitro BBB model and resulted in the improved performance of transgenic mice in spatial memory and recognition tests [197].

Cell-mediated transcytosis describes a phenomenon in which immune cells (e.g., macrophages, monocytes) are able to cross the BBB [198]. As such, immune cells can be exploited as a “Trojan horse”, and nanocarriers can be designed with the aim of their uptake into macrophages. This is then followed by the penetration of the immune cells (containing the nanocarrier) across the BBB [199,200]. Cell-mediated transcytosis has been less studied as a strategy to increase nanocarrier-mediated brain delivery; however, new applications of this approach are to be expected and could be promising for brain delivery of natural Nrf2 modulators. As the drug is encapsulated in the nanocarrier as well as taken up by immune cells, this represents a great possibility to decrease the rapid biotransformation of polyphenolic phytochemicals.

Among the many nanocarriers that have been proposed for improved brain delivery (e.g., polymeric nanoparticles, liposomes, micelles, dendrimers [201]), SLNs have been the most frequently studied and represent a promising strategy for the use of Nrf2 activators for the prevention and treatment of neurodegenerative diseases. SLNs have shown promise in several in vivo studies in animal models, where they can increase the bioavailability of different natural Nrf2 modulators in the brain compared to the free compounds. Particularly promising data were reported in a study by Kakkar et al., where curcumin-loaded SLNs were demonstrated to cross the BBB of rats. This provided the preferential distribution of curcumin to the brain and increased its brain bioavailability to 30-fold over that of the systemic circulation [142]. Successful SLN-mediated delivery to the brain has been reported in several other in vivo studies. These have included reports on increased brain bioavailability of Nrf2 modulators such as curcumin [138] and resveratrol [140,142]. Moreover, SLN-mediated improvements in the delivery of Nrf2 activators quercetin [202] and curcumin [138] to the brain even resulted in improvements in oxidative stress-related biochemical markers, such as glutathione levels, lipid peroxidation, and the enzymatic activities of superoxide dismutase, catalase, and mitochondrial complex. These three reports represent only a few of the many promising studies on the potential of Nrf2-activator-loaded SLNs not only for cancer prevention but also for the prevention and treatment of neurodegenerative diseases. As for nanoparticles, following intravenous administration, micelles can improve the brain delivery of natural Nrf2 modulators for the prevention and/or treatment of neurodegenerative disorders. Mixed micelles have been reported to improve the distribution of *trans-*resveratrol and to increase its brain bioavailability in rats [203]. Brain delivery can also be improved via the surface modification of nanocarriers. The attachment of various ligands to nanoparticles can enhance their transport across biological barriers, which is usually achieved via receptor-mediated endocytosis. Indeed, apolipoprotein E is known to bind to overexpressed LDL receptors on the BBB. SLNs containing resveratrol and surface modified with apolipoprotein E show enhanced permeability in in vitro BBB models [142].

## 6. Clinical Studies of Nanoformulations with Curcumin

The clinical studies of natural polyphenolic Nrf2 modulators are plentiful; however, few nanoformulations have been evaluated. Particularly, curcumin has been studied extensively for its clinical efficacy in the prevention and treatment of a myriad of pathologies such as malignant diseases, central nervous system disorders, cardiovascular diseases, inflammatory diseases, skin conditions etc. [204]. The above sections of this paper discussed improvements in the oral bioavailability of poorly soluble Nrf2 modulators, citing in vivo studies on animal models and some rare clinical pharmacokinetic studies. Solid dispersions [205,206] and micelles [207] have been reported to increase the oral bioavailability of curcumin in humans. Nevertheless, the role of nanotechnological advances in the delivery of curcumin and other natural Nrf2 modulators has rarely been clinically studied beyond pharmacokinetic evaluation.

Nano-curcumin—a nanonised form of this phytochemical characterised by submicron curcumin particle size—has elicited the most interest for its oral use in the clinical setting and has been evaluated for its activity against inflammatory pathologies. Nrf2 is mainly related to the modulation of oxidative stress; however, the implication of this pathway in inflammation has also been shown in the knockout of Nrf2 in mice, where it resulted in increased expression of pro-inflammatory cytokines and inducible nitric oxide-synthase (iNOS) [208]. Clinically, nano-curcumin was evaluated for its efficacy in inflammatory pathologies in four different studies. In three separate clinical studies, nano-curcumin decreased the levels of pro-inflammatory cytokines [209,210,211]. Lastly, Abdolahi et al. reported that nano-curcumin decreased the levels of inducible nitric oxide-synthase (iNOS) in serum and its expression in isolated peripheral mononuclear blood cells [212].

Liposomal curcumin is another type of clinically evaluated curcumin formulation. The efficacy of parenterally administered liposomal curcumin was evaluated in three separate clinical studies. A dose-escalation clinical study found that liposomal curcumin was shown to be tolerable for doses up to 300 mg/m^2^ in metastatic or locally advanced cancer patients [213]. Similarly, Storka et al. found that the infusion of liposomal curcumin was safe and tolerable for infusion at 120 mg/cm^2^ but induced transient changes in red blood cell morphology [214]. In this study, no comparison to non-encapsulated curcumin was given; however, results of a previously published in vitro study showed that empty liposomes could also affect red blood cell morphology [215]. This points to the importance of further clinical studies and more precise evaluation of the nanoDDSs such as liposomes. Lastly, a phase Ib clinical study by Bolger et al. reported on the potential interaction of curcumin with drugs targeting the renin-angiotensin system, which was further elucidated and confirmed in in vitro studies [216].

None of the described clinical studies pertaining to nano-curcumin and liposomal curcumin compared the efficacy or safety of the formulated curcumin to non-formulated curcumin, thereby preventing the comparison between nanoformulations and the native phytochemical. This is somewhat expected in orally administered poorly soluble Nrf2 modulators due to their extremely limited bioavailability. However, parenteral administration introduces the phytochemical as well as the entire nanoDDSs into the systemic circulation. Therefore, the evaluation of efficacy and safety of nanoformulations containing curcumin or other natural Nrf2 modulators would benefit greatly from comparative data on the administration of the non-formulated respective phytochemicals.

Few studies provide details of the actual effect of these phytochemicals on Nrf2 as the target. This is perhaps the greatest shortcoming of the current research protocols in the field of oral and non-oral use of natural Nrf2 modulators. For further reading on the topic, the reader is directed to a recent and comprehensive analysis that focused on clinical trials with dietary phytochemicals that target the activation of Nrf2 [16], although only 18 trials were eligible. Other problems with the currently available clinical data pertaining to Nrf2 activation include the high risk of bias and the different choices of phytochemicals and investigated tissues (e.g., peripheral blood, skeletal muscle, skin). Although the activation of Nrf2 was mainly evaluated directly as gene expression, few studies provided data on the efficacy of target activation, such as increases in Nrf2 protein levels or in the binding of Nrf2 to DNA [16]. It would be beneficial to plan further clinical investigations that combine the determination of several genetic and biochemical parameters. Therefore, the detailed implication of Nrf2 and its modulators in the clinical outcomes of oxidative stress-related diseases has yet to be elucidated.

## 7. Concluding Remarks

Due to the unfavourable solubility and pharmacokinetic properties of most natural polyphenolic Nrf2 modulators, as indicated above, drug delivery design approaches are vital to mitigate the issues of suboptimal bioavailability and rapid elimination. The increasing focus on research into oxidative stress-related diseases and their molecular mechanisms has further stimulated attempts to improve the delivery of natural Nrf2 modulators. The literature on the delivery of these compounds is extensive; however, Nrf2 inhibitors have received much less attention than Nrf2 activators. Among the activators, curcumin and resveratrol are the most highly studied to date. The resulting published literature includes a large number of studies focusing on several aspects of the delivery of Nrf2 modulators.

The mitigation of poor oral bioavailability is essential because of the properties of many natural Nrf2 modulators, such as poor solubility and stability, as well as extensive pre-systemic metabolism and rapid elimination. The current research trends indicate that improvements in oral bioavailability of natural Nrf2 modulators are largely based on the mitigation of their poor aqueous solubility that has been achieved through several formulation approaches (see Section 5). Among these, solid dispersions, SMEDDSs, and several types of nanoDDSs have been the most commonly described, and as such, are summarised in Table 2 and Table 3.

Successful oral bioavailability improvements based on increased solubilities and dissolution rates have been shown for several natural Nrf2 modulators in studies on in vivo animal models. Rodent models have usually been used here, and most formulations have achieved up to 10-fold increases in oral bioavailability, as shown by comparisons of the AUCs of plasma concentrations (i.e., non-formulated vs. formulated Nrf2 modulators). Only a few formulations have shown larger increases in AUC and c_max_ (up to 100-fold). However, a clear estimation of the increases in AUC and c_max_ is impeded by the non-standardised parameters across in vivo studies in animal models (e.g., different doses, different modes of enteral administration, blood plasma sampling times). This can also be seen by the variability in the plasma concentrations in animal models in terms of the oral administration of non-formulated and formulated curcumin provided in Table 4.

So far, only a handful of pharmacokinetics studies on different formulation approaches for oral delivery of natural Nrf2 modulators have been carried out in humans. In particular, the evaluation of the clinical translatability of nanoDDSs (e.g., nanocrystals, polymeric nanoparticles, SLNs, micelles) is falling far behind the broad selection of preclinical pharmacokinetics studies that have been carried out. Therefore, reports on these nanotechnological approaches for the oral delivery of natural Nrf2 modulators are particularly deficient in clinical data. An exception here is the clinical study on curcumin-loaded Tween 80 micelles, which have shown an impressive 185-fold increase in oral bioavailability [132]. These shortcomings of the currently available preclinical and clinical pharmacokinetics data do not allow any strong conclusions to be drawn on the most successful of the investigated formulations. Thus, it is also difficult to make any prediction for the most promising composition and formulation approaches for clinical testing of the natural Nrf2 modulators to promote their translation into practice. While it is evident that the more conventional approaches for oral delivery of Nrf2 modulators (e.g., solid dispersions) have the advantage of currently being better established and already under manufacture industrially, it remains unclear whether future clinical studies might shift this focus towards the use of nanoDDSs.

Due to the lack of clinical data, the effects of plasma concentrations of specific Nrf2 modulators on intracellular in vivo target activation have yet to be clarified. Therefore, the level of oral bioavailability required for optimised therapeutic outcomes is currently unclear, and thus there remains an undefined quantitative value for the improved pharmacokinetics needed to adequately benefit from the oral delivery of these compounds. Among the numerous published sources, there are also different approaches to the selection of natural Nrf2 modulators for incorporation into drug delivery systems. For example, many Nrf2 activators have been studied for cancer prevention as well as cancer treatment, even though it is known that the activation of Nrf2 may offer protective action to the cancer cell, thereby desensitising it to chemotherapy. Combined with the lack of clinical data, this necessitates a more precise and critical approach to the research of natural Nrf2 modulators and the evaluation of their potential.

NanoDDSs show great potential for the improved delivery of Nrf2 modulators parenterally, and hence for Nrf2 modulation in neurodegenerative diseases, which has encouraged research into their brain delivery. However, in vivo improvements in brain delivery have never been tested outside of animal models, and therefore present a lack of clinical data. Our current knowledge on nanoDDSs for improved brain delivery in humans has even larger shortcomings than those related to the lack of clinical data of oral administration.

Complete evaluation of nanoDDSs for delivery of Nrf2 modulators would also require the determination of the biological effects of empty nanocarriers on the Nrf2 target itself. However, the activation of Nrf2 by frequently studied polymeric and solid lipid nanoparticles has never been determined. It is of interest to elucidate the potential implication of empty nanoDDSs in Nrf2-related pathways.

Many polyphenols have been recognised for potential use in the field of oxidative stress-related diseases in terms of their modulatory actions on Nrf2. Based on the improved oral bioavailability of Nrf2 activators and consequently, on increased plasma concentrations, increased activation of Nrf2 would be expected. However, the relationship between the concentrations of these activators and the target responses measured need to be defined in clinical studies. Therefore, it is clear that despite the numerous studies on natural Nrf2 modulators, much work will still be needed on the biological evaluation of oxidative stress markers. This is imperative in order to gather the knowledge needed to optimize the technological approaches for the successful delivery of natural Nrf2 modulators.

## Figures and Tables

**Figure 1 pharmaceutics-13-02137-f001:**
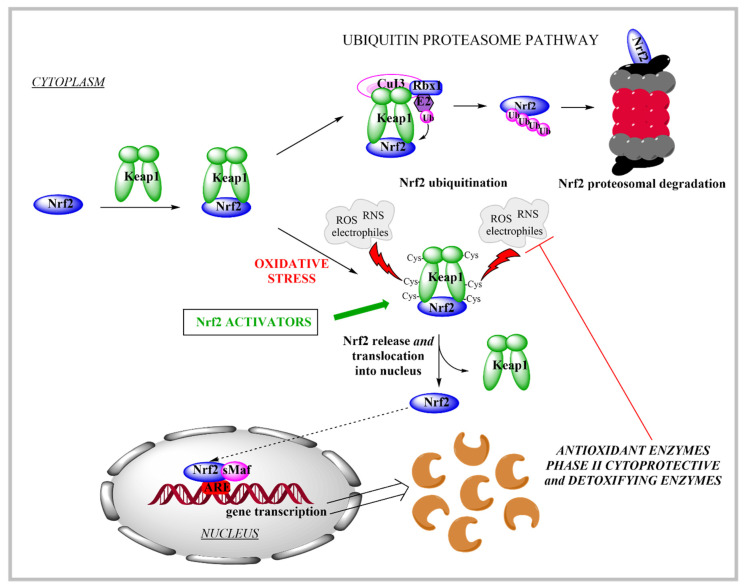
Schematic presentation of the Nrf2/Keap1 signalling pathway and Nrf2 responses to oxidative stress. Nrf2, nuclear factor erythroid 2-related factor 2; Keap1, Kelch ECH-associating protein 1; CuI3, Cullin3; Rbx1, Ring Box1; Ub, ubiquitin; ARE, antioxidant response element; sMaf, small musculoaponeurotic fibrosarcoma proteins.

**Figure 2 pharmaceutics-13-02137-f002:**
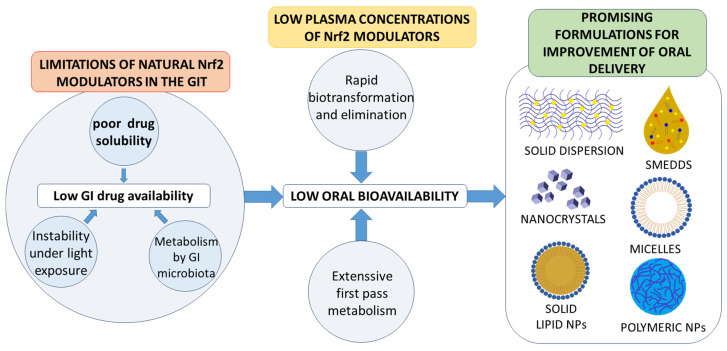
Limitations of natural Nrf2 modulators for oral use and common formulation strategies for mitigation of low oral bioavailability. GI(T), gastrointestinal (tract).

**Table 1 pharmaceutics-13-02137-t001:** Main characteristics of the frequently studied natural Nrf2 modulators for oral delivery. Numbers in parenthesis indicate the relevant references.

Name	Structure	BCS Classification	Aqueous Solubility (mg/L)	Log P	Mechanism of Action
**Curcumin**	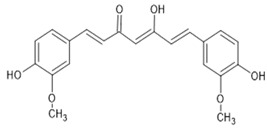	II	0.6 [44]	3.92 [45]	Nrf2 activation
**Genistein**	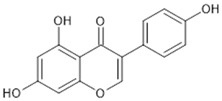	II	1.4 [46]	3.04 [47]	Nrf2 activation
**Luteolin**	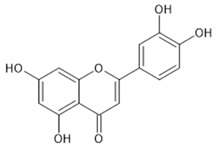	II	50.6 [48]	3.22 [47]	Nrf2 inhibition
**Kaempferol**	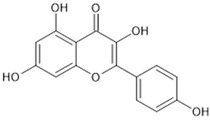	II	113.0 [49]	3.11 [47]	Nrf2 activation
**Quercetin**	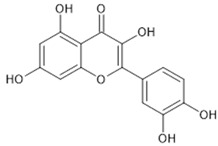	II	2.2 [50]	1.82 [49]	Nrf2 activation
**Resveratrol**	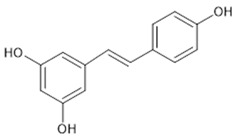	II	50.0 [51]	3.1 [52]	Nrf2 activation

**Table 2 pharmaceutics-13-02137-t002:** Examples of improved bioavailability of poorly soluble natural Nrf2 modulators achieved using different formulation approaches. Unless otherwise indicated, all pharmacokinetics studies were carried out following enteric modes of administration.

Natural Nrf2 Modulator	Delivery System	Experimental System	Improvements (Over Non-Formulated Compound)	Ref.
**Solid dispersions**				
**Curcumin**	TPGS and mannitol-based amorphous	In vivo PKs (rats)	65-fold AUC, 86-fold c_max_ increases	[71]
	HPMC-based amorphous	In vivo PKs (rats)	17.3-fold AUC, 62.8-fold c_max_ increases	[87]
	Arabinogalactan-based SD with decreased degree of curcumin crystallinity	In vivo PKs (rats)	7.85-fold AUC, 4.74 c_max_ increases	[88]
**Kaempferol**	Poloxamer 407-based amorphous	Solubility study	Increased solubility	
		Dissolution test	Increased dissolution rate	[93]
		In vivo PKs (rats)	1.9-fold AUC, 2.2-fold c_max_ increases	
**Quercetin**	Quercetin Phytosome (lecithin and quercetin complex-based formulation)	In vivo PKs (humans)	20.1-fold AUC, 20.4-fold c_max_ increases	[94]
**Resveratrol**	Soluplus and poloxamer 407-based	Dissolution test	Increased dissolution rate (Soluplus as optimal excipient)	
		In vitro permeability (Caco-2 cells)	Increased permeability with poloxamer	[91]
		In vivo (rats)	Increase bioavailability	
	Formulations based on different polymers: Eudragit E/HCl, Eudragit E PO, PVP K30, PVP VA 64, HPMC 6cp, HPC L-based, tested for dissolution; HPMC, Eudragit E/HCl solid dispersions, tested for PKs in vivo	Dissolution test	Increased dissolution rate, formation of supersaturated resveratrol solution; optimal performance for Eudragit-based solid dispersion	[90]
		In vivo PKs (rats)	4.2-fold AUC, 5.5-fold c_max_ increases; best performance for Eudragit solid dispersion	
**Self-microemulsifying drug delivery systems (** **SMEDDSs)**				
**Curcumin**	Formulation with added Eudragit E PO as precipitation inhibitor	In vivo PK (rabbits)	43.7-fold AUC, 30.7-fold c_max_ increases	[95]
	Emulsifier OP, Cremorphor EL, PEG and ethyl oleate-based	In vitro dissolution	Increased dissolution rate, solubility	
		Ex vivo absorption (rat intestine)	Increased absorption	[96]
		In vivo PKs (mice)	Increased bioavailability	
	Cremophor RH 40, Transcutol P and ethyl oleate-based	In vivo PKs (rabbits)	2.9-fold AUC, 2-fold c_max_ increases	[97]
	Ethanol, Cremophor RH40 isopropyl myristate-based	Dissolution study	Increased dissolution rate	[98]
		In vivo PKs (mice)	12.7-fold AUC, 3.1-fold c_max_ increases	
**Resveratrol**	Capryol 90, Cremophor EL, and Tween 20-based SNEDDS	In vivo PKs (rats)	3.2-fold AUC, 2.3-fold c_max_ increases	[99]

PKs, pharmacokinetics; AUC, area under the curve; c_max_, maximal plasma concentration; SNEDDS, self-nanoemulsifying drug delivery system; TPGS, D-α-tocopheryl polyethene glycol 1000 succinate; HPMC, hydroxypropyl methylcellulose; PVP, polyvinylpyrrolidone; HPC, hydroxypropyl cellulose; PEG, polyethene glycol.

**Table 3 pharmaceutics-13-02137-t003:** Examples of improved bioavailability of poorly soluble natural Nrf2 modulators achieved using different nanotechnological approaches. Unless otherwise indicated, all pharmacokinetics studies were carried out following enteric modes of administration.

Natural Nrf2 Modulator	Nano Drug Delivery System	Experimental System	Improvements (Over Non-Formulated Compound)	Ref.
**Polymeric nanoparticles**				
**Curcumin**	PLGA	In vivo PKs (rats)	15.6-fold AUC, 2.9-fold c_max_ increases	[125]
	PEG-PLGA		55-fold AUC, 7.3-fold c_max_ increases	
	PLGA	In vivo PKs (rats)	26-fold AUC, 7.2-fold c_max_ increases (adjusted for dose); increased t_max_, broader PKs profile, indicating prolonged release	[129]
	PLGA	In vivo PKs (rats)	5.6-fold AUC, 4.4-fold c_max_ increases	[115]
	G4 PAMAM dendrimer-palmitic acid core-shell	In vivo PK (mice)	2.1-fold AUC, 2.4-fold c_max_ increases	[128]
		In vivo memory and antistress (mice)	Improved efficacy of curcumin in formulation	
	Chitosan-pectinate	In vivo PKs (rats)	4.2-fold AUC, 1.4-fold c_max_ increases	[130]
	PLGA	In vitro solubilisation in dispersion of mixed micelles comprised of phosphatidylcholine and bile salts in vitro	Approximately 3-fold higher micelle-mediated solubilisation of NP-encapsulated curcumin compared to free curcumin	[131]
		In vivo PKs (rats)	Increased oral bioavailability (AUC) and c_max_	
**Resveratrol**	Eudragit RL 100	In PKs (rats)	7.25-fold AUC, 1.3-fold c_max_ increases	[132]
	Carboxymethyl chitosan	In vitro release	Increased dissolution rate, increased cumulative percent released drug	[133]
		In vivo PKs (rats)	3.5-fold AUC, 1.2-fold c_max_ increases	
	Galactosylated, non-galactosylated PLGA	In vitro release	Increased dissolution rate	[78]
		In vivo PKs (rats)	3.4-fold AUC, 4.3-fold c_max_ increases (galactosylated); 1.7-fold AUC, 2.4-fold c_max_ increases (non-galactosylated)	
**Quercetin**	PLGA	In vitro release	Improved release kinetics	[134]
**Solid lipid nanoparticles**				
**Curcumin**	Glyceryl behenate	In vivo PKs (rats)	39.1-fold AUC, 48.9-fold c_max_ increases	[135,136]
	Glyceryl behenate	In vivo PKs, biodistrib. (mice, rats)	Increased distribution into the brain	[137,138]
	Solid lipid nanoparticles with addition of P-gp inhibitors Brij78 and D-α-tocopheryl poly(ethene glycol) succinate 1000	In vitro permeability	Increased permeability (P_eff_)	[139]
		In vivo PKs (rats)	9.54-fold AUC, 3.54-fold c_max_ increases	
**Resveratrol**	Glyceryl behenate	In vivo PKs (rats; intraperitoneal)	Increased brain bioavailability	[140]
	N-trimethyl chitosan-g-palmitic acid surface-modified	In vitro release studies	Increased dissolution rate	[126]
		In vivo PKs (rats)	3.8-fold AUC, 1.6-fold c_max_ increases	
	Stearic acid	In vivo PKs (rats)	8.0-fold AUC, 1.6-fold c_max_ increases	[141]
	Apolipoprotein E-surface modified cetyl palmitate	In vitro blood–brain barrier models (hCMEC/D3 cell monolayers)	Enhanced permeability	[142]
**Micelles**				
**Curcumin**	Tween 80 micelles	In vivo PKs (humans)	185-fold AUC, 453-fold c_max_ increases overall (277-fold and 114-fold AUC, 806-fold and 251-fold c_max_ increases in women and men, respectively) (mean c_max_, 3228 ng/mL)	[143]
	Galactosamine-modified PEG-PLA micelles	In vitro release	Sustained release	[117]
		In vivo PKs (rats)	Galactosamine modification of micelles increases curcumin bioavailability (c_max_/D 5 ng/mL/mg/kg; no quantitative comparison to administration of non-formulated crude curcumin given)	
	Solutol HS15 +TPGS, Solutol HS15+Pluronic 127 mixed micelles	In vitro release	Sustained release	[127]
		In vitro permeability (Caco-2 cells)	Increased permeability, decreased efflux	
		In vivo PKs (rats)	6.2-fold and 5.7-fold AUC, 7.7-fold and 5.6-fold c_max_ increases (+TPGS, +Pluronic 127, respectively)	
**Genistein**	Pluronic F127 micelles	In vitro release	Sustained release	[144]
		In vivo PKs (rats)	Approx. 5-fold AUC, 4.7-fold c_max_ increases	
**Resveratrol**	Tween 80, Tween 20, medium chain triacylgly. mixed micelles	In vivo PKs (humans)	5.0-fold AUC, 10.6-fold c_max_ increases	[74]
	Poloxamer 407, TPGS mixed micelles	In vivo PKs (rats)	2-fold AUC, 2.8- and c_max_ increases	[145]
	Poloxamer 407, TPGS mixed micelles, loaded additionally with piperine (absorption enhancer)		5.7-fold AUC, 5.0-fold c_max_ increases	
**Quercetin**	Sodium taurocholate, Pluronic P123 mixed micelles (12.6% drug loading)	In vivo PKs (rats)	1.6-fold AUC, 1.8-fold c_max_ increases	[146]
	Methoxy-PEG-b-PLA	In vivo PKs (rats)	9-fold AUC, 3.1-fold c_max_ increases	[147]

PKs, pharmacokinetics; AUC, area under the curve; c_max_, maximal plasma concentration; PLGA, poly(lactic-*co*-glycolic acid); PEG, polyethylene glycol; PAMAM, poly (amidoamine); TPGS, d-α-tocopheryl polyethylene glycol 1000 succinate; PLA, poly(L-lactic acid).

**Table 4 pharmaceutics-13-02137-t004:** Comparison of maximum curcumin plasma concentrations (c_max_) achieved in animal models after administration of non-formulated and formulated curcumin. All formulations were administered via enteric routes. Unless otherwise stated, all c_max_ values are normalised with the dose administered.

Delivery System	c_max_/D of Curcumin (ng/mL/mg/kg)		Ref.
	Non-Formulated	Formulated	
**Solid dispersion**			
TPGS and mannitol-based amorphous	0.09	7.77	[71]
HPMC-based amorphous	1.3	82.7	[87]
Arabinogalactan-based, with decreased curcumin crystallinity	8.6	41	[88]
**Self-microemulsifying drug delivery systems**			
Ethanol, Cremophor RH40 isopropyl myristate-based	319	983	[98]
Cremophor RH 40, Transcutol P and ethyl oleate-based	80	162	[97]
**Polymeric nanoparticles**			
PLGA and PEG-PLGA	0.081	0.24 (PEG); 0.60 (PEG-PLGA)	[125]
PLGA	0.36	2.6	[129]
PLGA	16	68	[115]
G4 PAMAM dendrimer-palmitic acid core-shell	6.0	14	[128]
Chitosan-pectinate	70.5	100	[130]
**Solid lipid nanoparticles**			
Glyceryl behenate	5.8	286	[135]
With addition of P-gp inhibitors Brij78, TPGS	42	150	[139]
**Polymeric micelles**			
Tween 80 micelles (evaluated in clinical pharmacokinetics study)	2.6 ng/mL (average across both genders)	1189 ng/mL (average across both genders)	[143]
Galactosamine-modified PEG-PLGA micelles	/	5.0	[117]
Solutol HS15 +TPGS, Solutol HS15+Pluronic 127 mixed micelles	6.0	46 (for more successful TPGS-based formulation)	[127]

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
