# Peer review of "Drug Delivery Strategies for Curcumin and Other Natural Nrf2 Modulators of Oxidative Stress-Related Diseases"

_pharmaceutics, 2021, doi:10.3390/pharmaceutics13122137_

Round 1
Reviewer 1 Report
This review paper is well written and deserves publication after minor revisions.
The authors should define Nrf2 as nuclear erythroid-related factor 2 when used for the first time.
In section 4, the authors should cite innovative techniques used to improve active principles bioavailability based on the use of supercritical carbon dioxide. For example, quercetin and rutin have been coprecipitated with PVP in 10.1016/j.cep.2019.107689 and curcumin formed inclusion complexes with HP-β-CD in 10.3390/chemengineering5030059.
Author Response
Responses to reviewer number 1
We would like to thank you for the evaluation as well as useful comments and recommendations for the additions to the manuscript. We have revised the manuscript accordingly and the changes are seen and tracked in the document. The answers to the reviewer’s comments and suggestions are included below (the reviewer’s comments are boldfaced, while our responses are given in ordinary characters).
1.) This review paper is well written and deserves publication after minor revisions.
The authors should define Nrf2 as nuclear erythroid-related factor 2 when used for the first time.
Authors’ reply: We have included the definition of Nrf2 when used for the first time in the abstract (line 13), as well as in the text at the location of the first use of the term in the introduction of the paper (lines 46-47).
2.) In section 4, the authors should cite innovative techniques used to improve active principles bioavailability based on the use of supercritical carbon dioxide. For example, quercetin and rutin have been coprecipitated with PVP in 10.1016/j.cep.2019.107689 and curcumin formed inclusion complexes with HP-β-CD in 10.3390/chemengineering5030059.
Author’s reply: We thank the reviewer for this kindly provided recommendation of examples to include into the paper. As examples of innovative techniques based on the use of supercritical carbon dioxide, we have added a brief description of formulations with polyphenolic compounds prepared with the techniques of supercritical fluids and exhibiting increased dissolution rates of the drug were included on page 9 of the paper along with the 2 recommended references (lines 299-312).
Reviewer 2 Report
Dear Authors:
In the manuscript by Grilc et al has summarized the drug delivery strategies for curcumin and other natural Nrf2 modulators of oxidative stress-related diseases. I have just a few suggestion.
- Background and citations about ROS are not enough or missing in introduction. ROS plays an important role in cancer development, neurological disorders and so on. Please add more information about it. (Please cite: 1. Chen et al. Semin Cancer Biol. 2020 Oct 6:S1044-579X(20)30203-0. doi: 10.1016/j.semcancer.2020.09.012. 2. Shekhar et al. International Journal of Molecular Sciences. 2021; 22(4):2074. https://doi.org/10.3390/ijms22042074)
- In Page 2, Line 90-94:"Recently, research into cancer nanotechnology has also defined passive targeting approaches, such as the delivery of nanoparticles in angiogenesis as a compensatory mechanism of diffusion in cancers. Passive targeting depends upon the properties of the nanoparticles, such as their size, shape, surface properties and mechanical stiffness. Passive
targeting also involves the use of other innate characteristics of nanoparticles that can affect tumour targeting." Please give more references and experiments or research about nanparticle application in cancer therapy.
Author Response
Responses to reviewer number 2
We would like to thank you for the evaluation as well as useful comments and recommendations for the additions to the manuscript. We have revised the manuscript accordingly and the changes are seen and tracked in the document. The answers to the reviewer’s comments and suggestions are included below (the reviewer’s comments are boldfaced, while our responses are given in ordinary characters).
1.) Background and citations about ROS are not enough or missing in introduction. ROS plays an important role in cancer development, neurological disorders and so on. Please add more information about it. (Please cite: 1. Chen et al. Semin Cancer Biol. 2020 Oct 6:S1044-579X(20)30203-0. doi: 10.1016/j.semcancer.2020.09.012. 2. Shekhar et al. International Journal of Molecular Sciences. 2021; 22(4):2074. https://doi.org/10.3390/ijms22042074)
Authors’ reply: We have included a brief explanation about the implication of ROS in different pathologies in the introduction (lines 48-58). We thank the reviewer for recommending some additional citations. They were included in the paper.
In Page 2, Line 90-94:"Recently, research into cancer nanotechnology has also defined passive targeting approaches, such as the delivery of nanoparticles in angiogenesis as a compensatory mechanism of diffusion in cancers. Passive targeting depends upon the properties of the nanoparticles, such as their size, shape, surface properties and mechanical stiffness. Passive targeting also involves the use of other innate characteristics of nanoparticles that can affect tumour targeting." Please give more references and experiments or research about nanparticle application in cancer therapy.
Authors’ reply: We have included further brief general information about the nanotechnological approaches for the treatment of cancer (lines 104-106 and 109-117). Some more references were included to provide the reader with a basic understanding of the importance of nanotechnology and its potential for cancer therapy.
Reviewer 3 Report
This review provides an extensive update on the molecular mechanisms for Nrf2 modulators as well as the formulation strategies that can be employed to improve their oral PK profiles. The review is supported with 182 references from contemporary and older sources to provide a detailed and balanced discussion. While this is an extensively covered topic with respect to reviews, the inclusion of the section on nanotechnology approaches to improve Nrf2 modulator transport to the brain is not overly well covered. However, it would have been nice to see this section expanded upon by presenting and discussing future directions for brain-targeting transport of Nrf2 modulators. This could include reference to other work, not related to Nrf2 modulators, where brain delivery has been successful for specific bioactives with similar physicochemical attributes.
Further, it would have been nice to see a section on the clinical translatability of Nrf2 drug delivery strategies. There is an abundance of clinical studies that have been performed in this space, so I believe this review would significantly benefit from a section dedicated to clinical studies of Nrf2 nanotechnologies.
With that said, I believe that this manuscript is well-suited and worthy of publication in Pharmaceutics, following the above recommended changes.
Author Response
Responses to reviewer number 3
We would like to thank you for the evaluation as well as the kind and useful comments and recommendations for the additions to the manuscript. We have revised the manuscript accordingly and the changes are seen and tracked in the document. The answers to the reviewer’s comments and suggestions are included below (the reviewer’s comments are boldfaced, while our responses are given in ordinary characters).
1.) This review provides an extensive update on the molecular mechanisms for Nrf2 modulators as well as the formulation strategies that can be employed to improve their oral PK profiles. The review is supported with 182 references from contemporary and older sources to provide a detailed and balanced discussion.
While this is an extensively covered topic with respect to reviews, the inclusion of the section on nanotechnology approaches to improve Nrf2 modulator transport to the brain is not overly well covered. However, it would have been nice to see this section expanded upon by presenting and discussing future directions for brain-targeting transport of Nrf2 modulators. This could include reference to other work, not related to Nrf2 modulators, where brain delivery has been successful for specific bioactives with similar physicochemical attributes.
Authors’ answer: We have included a short expansion of the section covering the potential of brain delivery for the use of Nrf2 modulators in the treatment of neurodegenerative disorders on page (section 5 of the paper, lines 664-709). The basic approaches of modification of nanocarriers for the improvement of brain delivery were briefly described (exploitation of transporter-mediated transcytosis, adsorptive-mediated transcytosis, cell-mediated transcytosis). Some additional references were included to support this information. We included some examples which present the potential of these brain delivery strategies to be applied in the use of natural Nrf2 modulators.
2.) Further, it would have been nice to see a section on the clinical translatability of Nrf2 drug delivery strategies. There is an abundance of clinical studies that have been performed in this space, so I believe this review would significantly benefit from a section dedicated to clinical studies of Nrf2 nanotechnologies.
With that said, I believe that this manuscript is well-suited and worthy of publication in Pharmaceutics, following the above recommended changes.
Authors’ answer: We thank the reviewer for this suggestion and have included an additional section (lines 739-800), which briefly describes the clinical studies of nanotechnologies for the delivery of natural Nrf2 modulator curcumin. This modulator is the most widely studied among all polyphenolic compounds and also (the most) represented in clinical trials. We agree with the suggestion made by the reviewer that clinical studies about nanotechnologies for natural Nrf2 modulators should be included and have added a chapter discussing the current scope of clinical research in this field. Due to the fact, that the clinical translatability of nanotechnologies is currently still lagging far behind the clinical translatability of conventional dosage forms and delivery systems, only a small selection of clinical studies of nanotechnologies with curcumin were found during the literature review and searches in the PubMed database and ClinicalTrials.gov database (among those in the latter, many have not yet been concluded/carried out/reported on). The clinical studies were reviewed and shortly described in order to provide the reader with the understanding of the main trends of Nrf2 nanotechnologies that have made it to the clinical stage up to this moment.
Reviewer 4 Report
The review by Grilc et al summarizes the data related to the basic mechanistic background of the Nrf2 modulators and the limitations of their use.
Overall, the review article is fairly well written but the studies are not appropriately described. Only minor issues need to be addressed.
General suggestion:
Data reported in the in vivo studies must be well described. The description of the formulations is very detailed but the studies are not clearly showed.
Author Response
Responses to reviewer number 4
We would like to thank you for the evaluation as well as the kind and useful comments and recommendations for the additions to the manuscript. We have revised the manuscript accordingly and the changes are seen and tracked in the document. The answers to the reviewer’s comments and suggestions are included below (the reviewer’s comments are boldfaced, while our responses are given in ordinary characters).
1.) The review by Grilc et al summarizes the data related to the basic mechanistic background of the Nrf2 modulators and the limitations of their use.
Overall, the review article is fairly well written but the studies are not appropriately described. Only minor issues need to be addressed.
General suggestion:
Data reported in the in vivo studies must be well described. The description of the formulations is very detailed but the studies are not clearly showed.
Authors’ reply: We thank the reviewer for suggesting this addition. The many in vivo studies in animal models that were cited in the paper were described in an overview and this explanation was added to the paper in the beginning of section 4 (lines 299-312), before the detailed description of technologies use to increase oral bioavailability. We provided an explanation of the inclusion criteria for our studies – we reviewed studies, where animals (mice, rats or rabbits) were orally administered the studied Nrf2 modulators. Only studies with two separate groups were included – a group that received the native phytochemical and another group that received the phytochemical in the formulation (e.g. solid dispersion, nanoparticles etc.). The chosen animal models (rat, mice, rabbit) are included in the tables. We pointed out, that the showcased in vivo studies differed in administered dosages and sampling in order to remind the reader of the non-standardization between in vivo studies, which is also one of the main shortcomings of the currently available data covering the topic of this paper.
Round 2
Reviewer 2 Report
Authors made correction according to my previous suggestions. Strongly recommend for publishing.